# Roles of Drying, Size Reduction, and Blanching in Sustainable Extraction of Phenolics from Olive Leaves

Fereshteh Safarzadeh Markhali 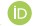

CEB—Centre of Biological Engineering, Campus of Gualtar, University of Minho, 4710-057 Braga, Portugal; id7987@alunos.uminho.pt or Fereshteh.safarzad@graduate.curtin.edu.au

**Abstract:** It is now known that olive leaves contain a sizable portion of polyphenols and there is much research highlighting that these natural ingredients favorably exhibit bio-functional activities. In this regard, many studies have focused on the exploration of optimum conditions involved directly in the extraction process. These investigations, while being highly valuable, may somewhat cast a shadow over other contributing factors such as those involved in the preprocessing of leaves, including size reduction, drying, and blanching. The use of these unit operations under appropriate conditions, together with other benefits, potentially exert improved surface area, homogeneity, and diffusion/mass transfer which may help develop the liberation of target bio-compounds. The research work in this area, particularly size reduction, is relatively limited. Although in various experiments they are incorporated, not many studies have focused on them as the main predictor variables. The performance of further research may help ascertain the magnitude of their effects. Consideration of the operational parameters in preprocessing step is equally important as those in the processing/extraction step that may comparably influence on the extraction efficiency. This review provides an overview of the potential roles of drying, size reduction, and blanching in the extraction efficiency of phenolics from olive leaves.

**Keywords:** preprocessing; size reduction; grinding; drying; blanching; olive leaves; by-products; phenolics; sustainable extraction

## 1. Introduction

Olive leaves form a large proportion of biomass residues derived from (i) agricultural practices (mainly at pruning stage) [1,2], and industrial processing of olive oil and table olives [3–5]. The abundance and bio-functional potential of the endogenous phytonutrients, such as polyphenols, in these residues, have progressively led to increasing global interest in developing extraction techniques to optimally re-use/re-direct these valuable natural components for high added-value applications. The emerging extraction technologies such as ultrasound-assisted extraction (UAE), microwave-assisted extraction (MAE), supercritical fluid extraction (SFE), etc., have shown potential for higher efficiency. Although the selection of extraction types/methodologies is greatly important, the achievement of sustainable processing rests decisively on the integration of associated factors before, during, and after extraction processing that may enable recovery of target biomaterials within an eco-friendly processing system. Among the key factors include the operational parameters associated with the preprocessing/pretreatment step which are comparably as important as those involved during the actual extraction. These include size reduction, drying, and blanching that are among the most common approaches applicable to olive leaves. They can be ideally incorporated into the processing flow, at the preliminary stage prior to the actual extraction process (Figure 1). Recent studies on the extraction of bioactive compounds from olive leaves observed improved extraction yield/bioactivity of the desired biomolecules using various extraction designs. In many experiments, olive leaves are typically preprocessed (e.g., dried and ground) but limited studies have examined

their effects on the extraction efficiency. Special attention often seems to be towards the processing conditions (as the main independent variables), which rather overshadows the significance of those involved in the preprocessing step (particularly, from the perspective of size reduction).

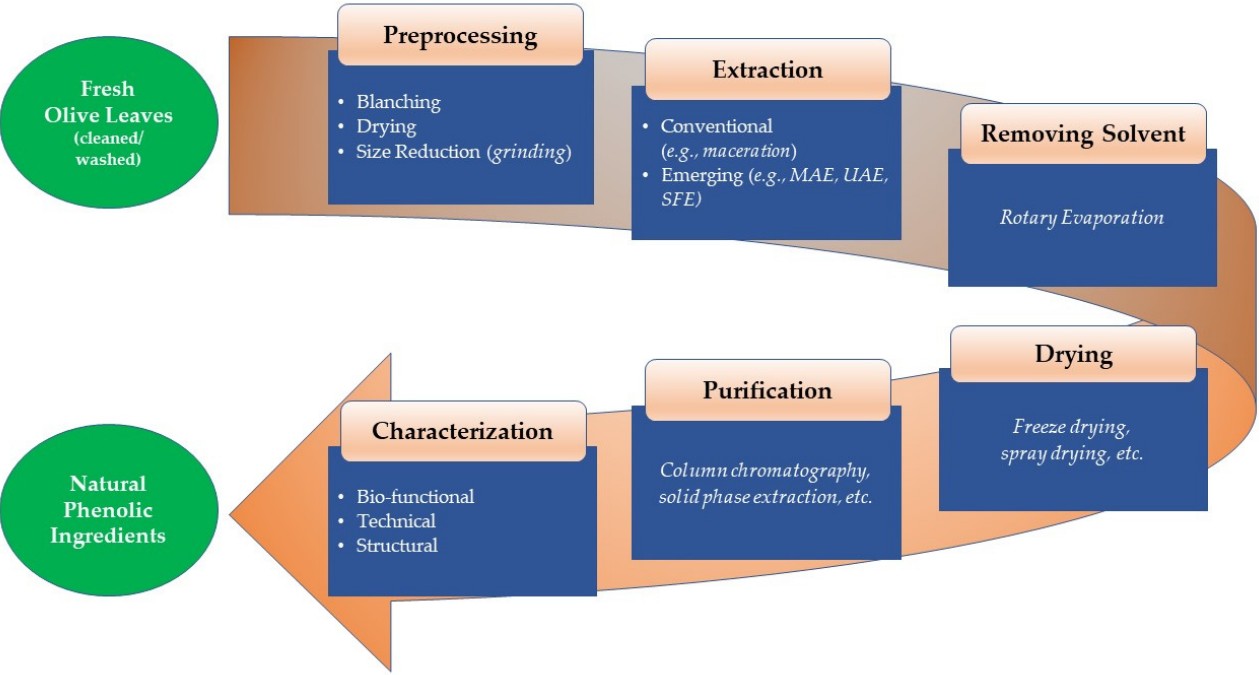

**Figure 1.** Typical steps associated with phenolic extraction from olive leaves.

The appropriate integration of preprocessing with processing means potentially reinforces the overall extraction efficiency in a sustainable manner. Depending on the selected types, they may significantly contribute to (i) reducing extraction time, (ii) increasing extraction yield and quality of nutritive molecules, and (iii) reducing input energy. For instance, particle size reduction, using dry grinding, which is often considered as an indispensable operation at the preliminary stage of exploitation of olive leaves, on its own plays a decisive role in the extraction efficiency, quantitatively and qualitatively. Other preprocessing means, drying and blanching, are comparably important and their operations may favorably/unfavorably affect the extractability and functionality of target biomolecules. This review discusses the roles of particle size reduction, drying, and blanching on the extraction efficiency of phenolic compounds from olive leaves. Given the fact that many studies tend to stress the importance of the main processing/extraction parameters, together with the potential practicalities of the abovementioned pretreatments, it may be worthwhile to highlight an overview of the effects of these unit operations and emphasize that there is a need to take them into account when addressing the challenges involved in the extraction/bioactivity of phenolic compounds from olive leaves.

## 2. Typical Preprocessing Means Applicable to Olive Leave Extraction

### 2.1. Drying

Drying fresh olive leaves at the preliminary stage prior to the extraction process is partly a deciding factor affecting the rate of extraction/bio-functionality of the released biomolecules. The dehydration of olive leaves may be achieved by different drying methods. Among the most common methods include (i) thermal drying through natural convection (such as shade and open sun drying), and forced convection (such as oven drying, solar drying, and heat pump drying) [6], and (ii) special forms of drying such as freeze-drying [6,7], microwave drying [6,7], infrared drying [7], and greenhouse drying [6]. The agro-industrial system often uses thermal energy for drying operations [8,9], which ex-

erts mainly two parallel phenomena: (i) heat flow from the driving force to the inside of the food, and (ii) mass transfer where moisture moves from the inside to the exterior/surface of the food and is then evaporated in the air [10–12], which enables the reduction in the food moisture content to a certain proportion [13–15]. The resulting moisture reduction may confer improved preservation and bioavailability of the endogenous phytonutrients including phenolic constituents via (i) protecting them against oxidative [16] and enzymatic activities [6], and spoilage microorganisms [17–19], (ii) enhancing the stability of large scale samples for further processing including extraction, (iii) enabling cellular destruction through rupturing the cell membrane that potentially gives rise to the liberation of bound phenolics [16,19]. The magnitude of the effects of drying on the extraction yield partly rests on the drying method and the type/physicochemical characteristics of the foods to be dehydrated. In the research of Nambiar et al. [20], the polyphenolic concentrations in fresh, air-dried, and oven-dried drumstick leaves represented around 141.59, 158.82, 185.32 mg/100 g, respectively.

Drying also finds applications for dehydrating the aqueous extracts to form them into powders (Figure 1) which partly enables addressing the inherent disadvantages of liquid extracts where the phenolic compounds are likely to be degraded by environmental conditions such as heat, and light [21]. In the research of Kiritsakis et al. [21], the aqueous extracts of olive leaves after UAE extraction were spray dried. The authors found that total phenols in dried powders were not adversely affected by drying operation (partly owing to the short drying time) and yielded around 6109 and 6985 μg/kg in Chalkidiki and Koroneiki, respectively. Further, Kashaninejad et al. [22] showed that freeze-drying of aqueous extracts of olive leaves is effective in the improved recovery of oleuropein (above 11% w/w) and luteolin-7-O-glucoside (1.4% w/w).

The incorporation of drying into the extraction process system, while being advantageous, may come with some challenges which entails the selection of appropriate drying method/conditions, and this partly relies on the extraction selectivity, together with other factors. Freeze-drying (lyophilization) is generally credited as being highly efficient in retaining the nutritive and sensory qualities of the dried products. Lyophilization, among other benefits, enables (i) minimal thermal damages to the food tissue, makes it a prime candidate for the protection of thermolabile compounds [23], and (ii) a porous structure that allows an increased penetration of the extraction solvents (if applicably used), and hence a greater chance of recovery of phytonutrients [19,24]. However, freeze-drying may come with challenges that may entail optimization to ensure the retention of endogenous phenolic constituents. Among the operational disadvantages includes the impact of ice crystals that is likely to cause damage to the cell structure of the food tissues [16]. Research demonstrates that olive leave extracts, pretreated with a hot air drier (120 °C), represented higher phenolic recovery when compared to those pretreated with a freeze dryer (loss of polyphenols reached up to 39% d.w.) [16]. On the other hand, in some research studies, freeze-drying has shown significant potential for the increased liberation of phenolics from the cell wall matrices. The research of Ghelichkhani et al. [25] demonstrated that freeze-dried olive leaves have great potential for extraction of total phenolic content (TPC), representing around 446.63 mg gallic acid equivalents (GAE)/g d.w. Martinho et al. [26] compared freeze-dried and non-dried (fresh) olive leaves and found polyphenolic concentrations in the range of 7.72–24.65, and 2.09–8.44 mg GAE/g leaves, respectively.

The temperature is among the prominent parameters in the drying operation that may favorably/unfavorably exert influence on the microstructure of the food (depending on the types and molecular characteristics of target bio-phenols). For example, research demonstrates that the use of hot air drying at a higher temperature (120 °C) has shown a better potency for the extraction of some phenolic compounds in olive leaves when compared to samples dried at 70 °C (through which the extraction yields of oleuropein and verbascoside decreased by 36% and 44%, respectively) [16]. This may be attributed to the effect of high temperature on the decrease in drying time, while the lower/moderate temperature may need lengthier drying time to accommodate a decrease in the moisture

content to reach the final/certain level in the dried products [16]. However, other studies suggest that heating may be unfavorable/less effective when the intention is to retain some bioactive compounds including oleuropein. Afaneh et al. [17] found more effective extractability of oleuropein when olive leaves were dried at room temperature (25 °C), yielding 10.0 mg/g dry olive leaves) that was greater than those obtained by other groups of leave samples (dried leaves at 50 °C and dry leaves harvested from the olive tree, yielding 1.7 and 2.5 mg/g of olive leaves d.w., respectively). These authors also indicated that drying operation is preferable due to its potential for improved oleuropein extraction as it was evidenced that fresh leaves contained negligible amounts of oleuropein (less than 0.1 mg/g). The fact that the extraction yield of oleuropein is generally greater in dried leaves compared to fresh ones is potentially due to the formation of oleuropein from oleuropein glucoside exerted by the enzymatic action of beta-glucosidase [17].

Further, Erbay and Icier [27] optimized the drying performance of a tray drier on olive leaves and reported that at a temperature of 51.16 °C, air velocity 1.01 m/s, and drying time of 298.68 min, the TPC loss in dried leaves reached up to 10.25% with a moisture content of 6.0%. Bahloul et al. [28] performed research on the roles of solar drying on the quality and functionality of olive leaves. In their research, it was found that the depletion of TPC (arising from the increase in drying time) is potentially minimized with a drying temperature of 50 °C and air velocity of 1 m/s. An optimized geothermal drying approach investigated by Helvaci et al. [29] described that drying at a temperature of 50 °C and air velocity of 1 m/s is effective in lessening the depletion of polyphenols in dried olive leaves. Nourhene et al. [30] investigated the drying kinetics of olive leaves from four Tunisian cultivars through solar drying (using an indirect forced convection) and reported that the rate of moisture diffusivity (ranging from $2.95 \times 10^{-10}$ to $3.60 \times 10^{-9}$ m$^2$/s) relies on drying temperatures as well as cultivar variations. The considerable effect of temperature variations during infrared drying on the yield recovery of polyphenols from olive leaves has been observed by Boudhrioua et al. [31]. Among the main findings of their research includes the extraction yield of the TPC in non-blanched leaves from Chemlali cultivar that increased with the rise in drying temperature (from 40 to 70 °C, representing 2.13 g, and 5.14 g/100 g d.w., respectively). The lowest concentration in the same cultivar was found in the non-dried/fresh leaves, 1.38 g/100 g of dry leaves.

The use of suitable drying technique may further help design a competing extraction system. Drying performance is partly interdependent on numerous deciding factors, among which include other complementary preprocessing operations. The drying process of olive leaves may be followed by grinding operation, and/or preceded by blanching process. In addition to the need for optimizing the parameters of each operation involved in preprocessing, the evaluation of their potential intercorrelations is equally significant particularly when considering a selective extraction of phenolics from olive leaves.

## 2.2. Size Reduction

Dried olive leaves are often subjected to dry grinding or the milling process which in part plays a crucial role in the resulting extraction efficiency. The main advantages of grinding are as follows: (i) intensified surface area via reducing the particle size that potentially improves the physicochemical and bio-functional activities [32], (ii) improved bulk density, and (iii) increased flow rate and porosity [33]. An ideal particle size reduction potentially enables improved dissolution and extractability which may in part address the poor solubility inherent in solvent extraction system [34].

In agri-food system the traditional grinders use milling operation based on a diverse range of devices including knife mills, disk mills, hammer mills, and ball mills [35]. Among the main disadvantages inherent in the conventional grinders include the possible increase in temperature (due to the friction and energy used to exert particle size reduction) and uneven distribution of particles [35]. The operating conditions, however, continue to be improved. For example, lately, ball milling has become more advanced,

using a temperature control system that makes it a viable milling approach applicable to thermolabile compounds [35,36].

As mentioned in Section 2.1, the process of drying may be followed by the operation of grinding. These two phenomena are both influential in the efficiency of further processing, and their appropriate operations may jointly complement the extraction system. Drying may give a further boost to grinding performance, and grinding, together with other benefits, helps increase the surface area which in part improves/reinforces the efficiency of further processing such as the drying performance involved after extraction (Figure 1) to powder aqueous extracts. Table 1 highlights a summary of the studies on phenolic extraction from olive leaves wherein drying and particle size reduction were employed prior to the extraction process (although mostly not selected as the predictor variables).

Much research has investigated the extractability and bioactivity of endogenous phenols from the perspective of solvent selection, in which aspects such as polarity, solubility, diffusivity, and non-toxicity were taken into consideration. The appropriateness of extraction solvents is of great importance as, together with other factors, the behavior of each phenolic group is different due to their variations in molecular and physicochemical properties. In this regard, the use of optimal size reduction may partly help complement the performance of extraction solvents. In the research of Stamatopoulos et al. [37], particle size (ranging 0.05, 0.1, 0.2, 0.315, and 1.0 mm) was assessed as one of the main factors to optimize a multistage extraction process of olive leaves (Table 1). It was found that the particle size reduction, up to about 0.315 mm, is highly effective in the exertion of phenolic liberation from dried olive leaves.

**Table 1.** Examples of drying and size reduction approaches used prior to phenolic extraction of olive leaves.

| Main Processing Factors for Phenolic Extraction | Drying & Size Reduction Prior to Extraction | Key Finding(s) | Reference |
|---|---|---|---|
| Leaves (pre-blanched) assessed as follows:<br>- Optimization via a single-stage extraction (particle size was among the key independent variables)<br>- Further optimized via multistage extraction system (compared to conventional method 40 °C, 48 h) | Drying: oven-dried with an air tray oven (60 °C, 4 h).<br>Size reduction: dry ground and sieved through 0.05, 0.1, 0.2, 0.315, and 1.0 mm. | - Single-stage extraction: Optimized conditions include 0.315 mm particle size, 70% ethanolic extraction, solid-liquid ratio of 1:7<br>- Multi-step extraction: Optimization with three stages (30 min, 85 °C) improved TPC (166.6 mg/g); Oleuropein (103.1 mg/g); luteolin-7-O-glucoside (33.7 mg/g); verbascoside (16.0 mg/g); apigenin-7-O-glucoside (13.8 mg/g).<br>- Multistage extraction enabled a 10-fold higher antioxidant activity compared to conventional extraction. | [37] |
| Steam blanching and hot water blanching (blanching time and particle size of fresh leaves accounted for the key parameters through optimization of blanching) | - For blanching optimization: particle size of fresh leaves ranged: above 20 mm, 20–11 mm, and 3–1 mm.<br>- For extraction: leaves (optimally blanched), air dried (60 °C for 4 h), and ground to 1 mm. | Optimized steam blanching (10 min, 20–11 mm particle size) improved oleuropein extraction (8.28 g/kg leaves d.w.), and antioxidant effects (4 to 13-fold increase, compared to those obtained from non-blanched ones). | [38] |
| Extraction solvents (methanol, ethanol, water, and acetone) | Drying: dried at room temperature in the dark.<br>Size reduction: dry ground to pass through a 20-mesh screen. | - Leaves extracted with 80% methanol exhibited higher TPC (392 mg GAE/g extract); total flavonoids (71 mg rutin equivalent/g); total tannins (18 mg GAE/g).<br>- Leaves extracted with ethanol (80%) exhibited DPPH antiradical activity (IC$_{50}$ = 1082.35 µg/mL). Total antioxidant activity (via linoleic acid system) was 76.36% with 2400 µg/mL extract. | [39] |
| Combining supercritical fluid extraction (with CO$_2$) and pressurized liquid extraction (PLE) | Drying: dried in the shade (ventilated).<br>Size reduction: dry ground to 3 mm particle size. | Oleuropein reached 10.44%, 9.5%, and 9.9%, with DPPH scavenging effects of 127.3, 145.3, and 138.6 µg/mL in defatted residues, using water (150 °C), ethanol (60%, 50 °C), and water (50 °C), respectively. | [40] |
| Freezing (conventional and liquid nitrogen) and drying (hot air drying and freeze drying) techniques | Drying: hot air-dried (70 °C for 50 min, 120 °C for 12 min) and freeze-dried.<br>Size reduction: dry ground to 0.05 mm particle size. | Using hot air drying (120 °C):<br>- Increased phenolics particularly oleuropein (108.6 mg/g d.w.).<br>- Antioxidant capacity via ferric reducing antioxidant power (FRAP) reached 109 mg Trolox equivalents (TE)/g d.w. | [16] |
| Optimization of MAE compared to conventional and UAE | Drying: dried at ventilated room temperature.<br>Size reduction: dry ground to pass through a 60-mesh. | Competitive effectiveness of MAE (5 min, 50% ethanol) in increasing TPC (76.6 mg GAE/g), and flavonoids (5.8 mg quercetin equivalent/g extract). | [41] |
| Hybrid extraction protocol (conventional ethanol extraction subsequent with supercritical fluid antisolvent extraction) | Olive leaves with 8% moisture content ground at room temperature to 1 mm particle size. | Concentrated yield of oleuropein powder reached up to 36% (35 °C, 150 bar). | [42] |

**Table 1.** *Cont.*

| Main Processing Factors for Phenolic Extraction | Drying & Size Reduction Prior to Extraction | Key Finding(s) | Reference |
|---|---|---|---|
| Optimization of aqueous extraction using water | Drying: dried at 120 °C for 90 min. Size reduction: dry ground to 0.1 mm. | Maximum TPC (32.4 mg GAE/g) yielded through extraction at 90 °C for 70 min, solid/solvent ratio of 1:60 g/mL Antioxidant capacity, using DPPH and FRAP, reached 85.26 and 91.03 mg TE/g, respectively. | [43] |
| Optimization of UAE | Drying: air-dried at 40 °C Size reduction: dry ground to a 0.5 mm | Increased yield of oleuropein (10.65%) using 50% acetone, 60 °C, 10 min. | [44] |
| Extraction methods (solvent extraction, UAE, and reduced pressure extraction) | Drying: dried at ambient temperature (no exposure to solar radiation). Size reduction: ground with a high-speed crusher to pass through a 40–60 mesh. | Increased oleuropein via combined UAE and reduced pressure extraction (92.3% extraction efficiency in a single run). | [45] |
| Olive leaves (dried and fresh) from different cultivars | Drying: freeze-dried Size reduction: ground to 0.1 mm | - TPC ranged 7.72–24.65 and 2.09–8.44 mg GAE/g in dried and fresh leaves, respectively. - Effective in inhibiting proliferation of human carcinoma cell line (e.g., freeze dried leaves ranged from 0.07 to 2.40 µg phenolic constituents/well). | [26] |
| - Extraction methods (MAE, Soxhlet) - Extraction solvents | Drying: open air-dried in the dark. Size reduction: ground and sieved (<2 mm) | Higher TPC (76.1 mg GAE/g), and antioxidant activity (78.0 mg TE/g) in Soxhlet extracted leaves (50% ethanol). Oleuropein was the key component. MAE was comparably effective. | [46] |
| Extraction methods (MAE, UAE, maceration) | Drying: oven-dried (24 h, 40 °C). Size reduction: ground to pass through a 60-mesh. | MAE extracts (86 °C, 3 min) exhibited higher TPC (104.22 mg GAE/g), with 90.03% antioxidant activity. | [47] |
| - Preprocessing leaves: drying, non-drying (fresh leaves) - Solvent variations | Drying: freeze-dried (−50 °C, 36 h, 0.08 mbar); hot air oven dried (120 °C, 8 min). Moisture content < 1%. Grinding: milled using a blender | - Hot air-dried leaves extracted by 30% ethanol exhibited highest TPC (151 mg/g d.w.), with DPPH-scavenging activity of 922 µmol TE/g. - The use of water (100%) comparably effects on increased TPC (144 mg/g) of hot air-dried leaves. | [48] |
| - Successive extraction techniques - Samples: Olive mill leaves and collected leaves from olive trees | Drying: air-dried Size reduction: ground to 1 mm particle size | - TPC in extracts from olive mill leaves: 4476–6167 mg GAE/100 g. - Extracts from Olive tree leaves (UAE prior to alkaline extraction) contained TPC around 13,108 mg GAE/100 g; oleuropein (12,694 mg/100 g); luteolin 7-O-glucoside 903 mg/100 g; with antioxidant efficiency of 59,651 µmol TE/100 g - Highest concentration of oleuropein in olive mill leaves was 1790 mg/100 g extract. | [49] |

**Table 1.** *Cont.*

| Main Processing Factors for Phenolic Extraction | Drying & Size Reduction Prior to Extraction | Key Finding(s) | Reference |
|---|---|---|---|
| Optimization of UAE extraction | Dried leaves were ground to 0.9−2.0 mm prior to extraction | - Extraction with 43.61% ethanol, 34.18 °C, 59 min exhibited increased TPC (43.825 mg GAE/g dried leaves). <br> - Total flavonoids (31.992 mg catechin equivalents/g dried leaves) through 70% ethanol, 34.44 °C, 60 min. <br> - DPPH inhibiting capacity ranged 89.3%–90.5% | [50] |
| - Extraction solvents (ethanol, methanol, acetone, and water) <br> -Extraction methods (MAE and maceration) | Drying: dried in the shade <br> Size reduction: ground to pass through a 60-mesh size screen | TPC using ethanol (50%) represented 88.298 and 69.027 mg GAE/g extract d.w. via MAE and maceration, respectively. | [51] |
| Pressurized liquid extraction using water and ethanol | Drying: dried at ambient condition (not exposed to solar radiation) for about 50 days (depending on relative humidity). <br> Size reduction: cryogenically ground using liquid nitrogen. | - TPC yielded 58.7 and 45.8 mg GAE/g, using water (200 °C) and ethanol (150 °C), respectively. <br> - Through water extraction, hydroxytyrosol was the principal phenolic component (up to 8.542 mg/g extract). Through ethanol extraction, oleuropein was the principal component (up to 6.156 mg/g extract). <br> - Extraction with water (200 °C), and ethanol (150 °C) showed effective DPPH scavenging activities ($EC_{50}$ = 18.6 and 27.4 µg/mL, respectively). | [52] |
| Solvent extraction (80% methanol) | Dried/micronized olive leaves (commercial powders) | - Extraction enabled TPC up to 131.7 mg GAE/g leaves d.w.), total flavonoids with 19.4 mg quercetin equivalents/g, and oleuropein 25.5 mg/g d.w. <br> - Antioxidant effects: 281.8 mg TE/g, and $EC_{50}$ 13.8 µg/mL using FRAP and DPPH, respectively. | [53] |
| Effect of drying on supercritical extracts | Drying: conveyer belt dryer (air temperatures range: 50, 60 and 70 °C; residence time: 180, 120 and 60 min). <br> Size reduction: ground with a knife mill for 5 min, and sieved (274 µm particle mean diameter). | Drying at 60 °C for 120 min presented higher TPC (36.1 mg GAE/g d.w.) in supercritical extracts, with 73% DPPH inhibiting activity, $EC_{50}$ = 1.1 µg/mL | [54] |
| - Microencapsulation of olive leaves <br> - Frying methods: starch gluten fried dough added with microencapsulated leaves | Drying: pre-blanched leaves dried in force air oven (at 45 °C for 18 h). <br> Grinding device: windmilled. | - Olive leaf extract: TPC was 25.7 mg GAE/mL extract; oleuropein was 28.4 mg/mL extract: $EC_{50}$ = 0.15 mg GAE/mL extract (DPPH) and 109 µmol TE/mL extract (FRAP). <br> - Highest TPC in atmospheric fried dough containing microencapsulated leaves. | [55] |

**Table 1.** *Cont.*

| Main Processing Factors for Phenolic Extraction | Drying & Size Reduction Prior to Extraction | Key Finding(s) | Reference |
|---|---|---|---|
| Olive leaf extract (80% ethanol) and fractions | Drying: dried at 40 ± 5 °C for 6 h Size reduction: ground to pass through a 20–30 mesh | - Ethanolic extract (80%) contained TPC (148 mg/g); total flavonoids (58 mg naringin equivalents/g); oleuropein (the main phenol, 102.11 mg/100 g). Rutin, vanillin, and caffeic acid (minor phenols) represented 1.38, 0.66, and 0.31 mg/100 g, respectively. <br> - Among the fractions: butanol fraction showed greatest antioxidant activity with highest TPC (175 mg/g), and flavonoids (75 mg/g). | [56] |
| Optimization via UAE extraction | Drying: air-dried at 25 °C for 7 days. Size reduction: coarsely ground using mortar and pestle. | Compared to maceration, oleuropein increased (30%) with UAE (70% ethanol, 25 °C, 2 h, solid: solvent ratio of 1:5). | [57] |
| Extraction kinetics and temperature with UAE and conventional | Dried in a tunnel microwave dryer (70 °C, 1200 W, 10 min) and ground prior to extraction. | - Oleuropein, TPC, and antioxidant capacity increased with the rise of temperature (through both UAE and conventional). <br> - Oleuropein ranged from 6.48 to 6.65 g/100 g d.w.) through UAE that enabled 88% oleuropein extraction in the 1st min. <br> - Using UAE at low temperature (10 °C) competitively exhibited higher oleuropein (5.71 g/100 g d.w.) in 10 min, compared to the conventional (5.15 g/100 g d.w.). | [58] |
| Drying of aqueous extracts (freeze-drying and spray-drying) | Leaves (after being washed) kept in the shade (48 h), and ground (80-mesh screen). | - Freeze-dried extracts: TPC (446.63 mg GAE/g d.w.), total flavonoids (298.16 mg quercetin/g), tannins (117.32 mg GAE/g), with 96.57% antioxidant activity. <br> - Spray-dried extracts: TPC (442.84 mg GAE/g d.w.), flavonoids (396.4 mg quercetin/g), tannins (128.71 mg GAE/g), with 96.05% antioxidant activity. | [25] |
| Optimization of extraction conditions including drying methods and solvent types/ratio | Drying methods: shade-drying; microwave (2450 MHZ, 80 sec); and vacuum (– 0.5 bar, 55 °C, 24 h). Size reduced by grinding. | - Microwave drying of fresh leaves provided the highest TPC (6.45 g GAE/100 g dried leaves). <br> - Favorable extraction conditions (40% ethanol 60 °C, 120 min) enabled high antioxidant activity ($IC_{50}$ = 18.92 μg/mL), with a TPC around 6.63 g/100 g. | [59] |

Among the advanced milling methods includes superfine grinding (micronization), which has shown to have great potential for increased surface area, uniform size reduction, bulk density, and flowability. This type of grinding, compared to the conventional methods, may exert greater physicochemical changes, dispersion, and solubility [60]. Particle size reduction, driven by superfine grinders are typically in the range of 100 μm to 0.001 μm [36]. Among the milling devices commonly used for micronization include jet mills, ball mills, vibration mills, agitated media, and roller mills [60]. Numerous studies demonstrate the significant potential of superfine grinding for improved bio-functional and physicochemical properties of phytonutrients in a broad range of foods, for example, olive pomace by-products [61]; red grape pomace [62]; pomegranate peels [63], rice bran [64], persimmon by-products [65], *Quercus salicina* Blume leaves [66], and ginseng species [67].

Research demonstrates that finely ground stevia leaves (with a particle size of 200 μm) facilitate high recovery of the target bioactive compounds during ultrasound extraction within a shorter time, which may signify the occurrence of mass transfer during grinding prior to the extraction process [68]. On the other hand, in the research of Chen et al. [69], there was no significant difference between flavonoid content in the superfine ground mulberry leaves and those obtained from coarsely ground leaves. This may be indicative of the ineffectiveness of the grinding temperature used (45 °C) on the extractability of these phenolics [69]. Some studies suggest that the micro-size pulverization may have adverse effects on the extraction rate and bioactivity of some functional nutrients. This in part depends on the nature of the food and the milling conditions, together with others. For instance, green tea represented reduced proportions of catechins, and total phenols when it was ground to a finer particle size [70]. Tchabo et al. [71], through their research to optimize phytonutrient properties of aqueous extracts of mulberry leaves, found that the particle size less than 2 μm may bring about oxidation, which partly signifies the exposure of the released biomolecules to the oxidizing agents.

In the case of olive leaves, there is limited information on micro-size particles and their effects on extraction efficiency. Lins et al. [53], through their research on antioxidant activity (*in vitro*), used dried micronized olive leaves (Table 1) extracted by 80% methanol (agitated for 170 rpm at 25 °C). Among the main findings of their study include the improved concentration of total phenolics (131.7 mg GAE/g leaves d.w.), oleuropein (25.5 mg/g leaves d.w.), and flavonoids (19.4 mg quercetin equivalents/g leaves d.w.). On the other hand, different from what is generally expected, the particle size reduction less than 0.2 mm is reportedly disadvantageous for olive leave extraction, causing a reduction in the extractability [37]. This may be attributed to the potential agglomeration of primary particles that adversely affect solvent permeation, and hence render the extraction difficult [37].

Cryogenic grinding, namely freezer grinding or cryomilling [36], is among the advanced grinding methods that allows finer particle size via grinding operation at extremely low temperature, often making use of liquid nitrogen [72,73]. Compared to traditional methods, it offers advantages, among which include enabling (i) very small, evenly distributed particle size, (ii) increased yield, (iii) economically viable milling operation, (iv) no thermal damage—makes use of a cryogenic mechanism and develops embrittled/fractured products [35,73,74], and (v) hindrance to the formation of oxide layer on the particle surface [36,75]. Numerous research studies performed investigations on the effect of cryogenic grinding on the extraction yield and bio-functional activity. Saxena et al. [74], found improved concentration of total phenolics in the cryogenically ground fenugreek seed extracts (ranging 75.72–94.03 mg GAE/g depending on the genotype variations) compared to the non-cryomilled samples. Sharma et al. [76] observed enhanced recovery of polyphenols from cryomilled ajwain seeds and the maximum recovery of TPC (in genotype samples extracted by dimethyl sulfoxide) was around 168.0 mg GAE/mL of crude seed extract. Saxena et al. [73], through their investigation on the quality and antioxidative profile of coriander, observed total phenolic recovery that, depending on the genotypes examined, ranged between 32.44 to 92.99 mg GAE/g crude seed extract of cryomilled samples.

Among the limited studies on cryogenic grinding of olive leaves includes the research of Herrero et al. [52], wherein olive leaves were cryomilled in advance of pressurized liquid extraction (PLE) using water (200 °C) and ethanol (150 °C) and yielded TPC around 58.7 and 45.8 mg GAE/g extract, respectively (Table 1). The principal components following extractions with water and ethanol were hydroxytyrosol and oleuropein, respectively. In another research study, the cryo-ground olive leaves, prior to subcritical water extraction at 200 °C, contained TPC of around 77.84 mg GAE/g extract [77]. The evaluations of these studies are significantly valuable; having said that, there is still uncertainty regarding the effects of the milling method (cryogenic) on the resulting extraction of each class of phenolics.

Selection of the milling operation for particle size reduction seems to play a decisive part in the quality and extractability of bio-compounds. The process of extraction kinetics, particularly through the washing phase and diffusion phase, is greatly reliant on the changes that occur in the cellular structure (disruption/intactness of the tissue cells). The rate of extraction during the washing process is potentially accelerated by the increased surface area of the food tissue. Khemakhem et al. [58] indicated that the enhanced extraction rate during the washing phase is partly attributed to the effects of the particle size reduction in olive leaves that were initially milled prior to the extraction process (Table 1).

### 2.3. Blanching

Blanching, as a thermal pretreatment, can be used at the initial point of olive leave extraction, often in advance of the drying operation. Blanching helps alter/weaken the cellular structure of food tissue and potentially enhance the overall extraction efficiency [78,79]. Among various thermal blanching are conventional (hot water blanching), steam blanching, ohmic heat blanching, microwave blanching, and superheated steam impingement blanching [80]. The potential effectiveness of blanching in promoting the extraction yield is partly indicative of inactivating polyphenol oxidase, the enzyme prominently responsible for the reduction/oxidation of bio-phenols [81]. The study by Zeitoun et al. [81] describes that the blanching of olive leaves (90–95 °C for 20 s), extracted with 70% ethanol, exerts effects on the liberation of total phenols (593.0 µg GAE/g, around 61.70% increase) which was also positively correlated with greater antioxidant activity.

Hot water blanching of olive leaves, as an efficient thermal pretreatment, has a potential for improved extraction of phenolics such as oleuropein that is reportedly 8-fold higher compared to that released from non-blanched leaves [38]. Sucharitha et al. [82] observed an improved extraction of oleuropein up to 35–38% from hot water-blanched olive leaves (50–70 °C for 10 to 30 min). However, blanching by hot water may cause leaching or dissolution of some phytonutrients (particularly hydrophilic compounds), together with other downsides inherent in generating wastewater and the need for increasing the drying time [38].

Steam blanching is generally considered a better alternative to hot water blanching. Stamatopoulos et al. [38], from their experiments, used steam blanch pretreatments on olive leaves for 10 min, followed by solvent extraction (70% ethanol) at 40 C for 30 min, and observed increased oleuropein extraction that reached up to 35-fold greater than the corresponding components from non-steam-blanched samples. The authors also investigated the particle size of fresh olive leaves as the main predictor variable to optimize steam blanching (Table 1) from which a higher amount of oleuropein (8.28 g/kg leaves d.w.) was obtained from the leaves with 20–11 mm particle size blanched for 10 min. This technique appears to have great potential. The performance of broader investigations to assess its correlation with other preprocessing factors (e.g., drying and size reduction) may help devise a viably scalable design that potentially (i) allows minimal loss of water-soluble molecules, and (ii) enables optimum efficiency during drying, via reducing the drying time and suppressing excessive break-up of the cell membrane that may otherwise render the moisture mobility difficult during the drying operation.

Blanching and drying may favorably/unfavorably affect the rate of phenolic extraction interdependently. Nobosse et al. [83] examined the influences of steam blanching and drying methods on the composition and bioactivities of phytonutrients in Moringa leaves. Their study suggested that drying (solar and electric drying) may have adverse effects on the loss of phenolics in both blanched and fresh leaves (the extent of loss was greater in blanched/dried samples). The blanched/undried samples contained TPC of around 3.40% while the blanched leaves/solar dried and blanched/electric dried represented 2.32% and 2.67%, respectively. Moreover, it was found that the fresh leaves (non-blanched/undried) represented TPC of around 3.28% which was rather close to those in blanched/undried ones.

A novel study patented by Musco et al. [84] describes the production of supreme quality olive leaf powder through infrared dry blanching (concurrent use of blanching and drying) to produce olive leave powders with an improved yield of soluble polyphenols and greater antioxidant capacity, applicable to various industrial (food/non-food) uses. Boudhrioua et al. [31], through their research, used blanching in advance of drying of four types of olive leave cultivars (Chetoui, Chemlali, Zarrazi, and Chemchali) and highlighted that the variability of phenolic concentrations across different cultivars is partly dependent on the temperature variations of drying (infrared). For instance, the increment of polyphenols at a lower drying temperature (40 °C) was achieved in all groups of leaves (excluding Chetoui). It was also reported that the non-blanched/dried samples (Chemlali cultivars) required less drying time, compared to the blanched/dried ones, to reach the final/desired moisture content.

Olive leaves may have bio-functional potential for enrichment of other types of foods, such as edible/vegetable oils. Research by Majetic Germek et al. [85] compared the effects of adding fresh and dry/steam blanched olive leaves to rapeseed oil. It was observed that the addition of fresh leaves showed potential for phenolic increment, while the dry/steamed blanched leaves had great ability to increase chlorophylls in the oil. Another study by Gonzáleza et al. [86] assessed phenolic properties from encapsulated olive leave extracts (using sodium alginate through spray drying), wherein the leaves were initially blanched at 95 °C for 4.5 min and dried at 45 °C prior to extraction with 50% ethanol. Although the blanching factor was not considered as the target predictor variable for their study, it was found that the encapsulated extracts had significant potential for bio-accessibility and bioavailability (58% and 20%, respectively).

## 3. Future Perspectives

A sizeable proportion of research has emphasized the importance of various parameters associated with phenolic extraction from olive leaves. There are numerous challenges involved to achieve a sustainable extraction. Among other factors is the importance of size reduction, drying, and blanching which have been somewhat less studied (as the main predictor variables) compared to other substantial factors. More extensive investigations can help determine the magnitude of their effectiveness in the extractability and quality of the desired components. There needs to be clear conclusions on the optimal conditions of particle size reduction, which may entail a comparative study between coarse particles and micronized particles, and to identify their effects (benefits or impacts) on the extraction.

It may also be reasonable to perform research based on an interface between drying and grinding, to determine the effect(s) of particle size reduction on the efficiency of further processing performance, e.g., the secondary drying employed after the extraction process (to dry aqueous extracts). Moreover, the choice of blanching approach is potentially dependent on the type of the selected drying, and vice versa. This can further annotate the reason for gaining further knowledge of the intercorrelations between various unit operations involved at the pre-extraction step. In addition, a deeper knowledge may be required from the perspective of determining the rate of phenolic content in pretreated leaves before the actual extraction process and comparing it with the corresponding compounds obtained after completion of the extraction process. This information may be of great value due to the possibility of a significant release of compounds on pretreated samples (without being

subjected to the extraction) that may be quantitatively comparable to the yield(s) after the extraction process.

Many research studies have found novel solutions to optimally valorize olive leaves through various emerging extraction means. One of the main factors considered in numerous experiments is the appropriateness of extraction solvent type/ratio which comes with a significant challenge. The use of water as an eco-friendly solvent is undoubtedly desirable, primarily due to its non/less environmental/health hazard, non-toxicity, and low expenditure [87,88]. Although using water for phenolic extraction may be less preferred due to its inherently weaker extraction efficiency when the extraction temperature is low [48], it is reportedly effective in the increased extraction of total/some phenolics at higher temperatures. For example, using water at 90 °C enhanced phenolic extraction from grape seeds [89], and increased recovery of oleuropein and hydroxytyrosol from olive leaves [90]. As shown in Table 1, the total phenolic content in hot air-dried olive leaves (120 °C) extracted by water was comparable to those by ethanol solvents [48]. In this respect, it may be of value to perform deeper research to interact the extraction ability of water solvents with the key parameters in preprocessing operations, including the temperature(s) selected for drying fresh leaves.

Future studies should potentially involve a broad range of evaluations on the valorization of olive leave residual biomass, among which may include an integration of preprocessing and processing that may partly help formulate a feasible processing system to deliver better extraction performance and greater sustainability for olive leave reutilization.

## 4. Conclusions

The extraction of bioactive compounds such as polyphenols from olive leaves has been largely investigated. In this regard, many research studies have focused on factors involved directly in the extraction process and their effects on the optimum extractability of natural nutrients from olive leaves. These studies, while being significantly valuable, seem to have relatively overshadowed the importance of the roles of preprocessing means in the overall extraction efficiency and quality of the final product. Drying and grinding are often considered vital unit operations and have been incorporated into many experiments of olive leave extraction, but there is limited information available in the literature specifying their influences on the extraction.

Preprocessing of olive leaves, depending on the selected extraction approach, may play a decisive role in the improvement of extraction efficiency. Among others, they exert improved surface area, mass transfer, and extraction yield/acceleration. Size reduction is a major preprocessing operation that requires further research to gain a deeper understanding of the effect(s) of grinding parameters. Drying and blanching, although have been the topic of some assessments, need further evaluations that can help provide a better understanding of the magnitude of their effects. Moreover, a comparison study before and after extraction can potentially help understand the extraction yield through the grinding (prior to the extraction stage) and after the extraction process. This may help reinforce the overall extraction system, regardless of the extraction method (conventional or emerging).

Given the limited information available on the effects of size reduction, drying, and blanching on the overall extraction of phenolics from olive leaves, together with the possibility that the extraction of bio-compounds may be significantly influenced by these operations, more extensive research studies are needed in this respect. The findings may partly contribute to the development of innovation patterns for the valorization of olive leaves wherein a more advanced processing system may enable sustainable extraction of bio-functional materials such as polyphenols from this residual biomass generated from both agricultural and industrial activities of olives.

**Funding:** This review received no external funding.

**Conflicts of Interest:** The author declares no conflict of interest.

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
