# Peer review of "Roles of Drying, Size Reduction, and Blanching in Sustainable Extraction of Phenolics from Olive Leaves"

_processes, doi:10.3390/pr9091662_

Round 1
Reviewer 1 Report
I find the paper is interesting to publish because it gives a good overview of the literature. This is a good starting point for optimizing the extraction condition of polyphenols from olive leaves. However, I think that one part needs to be strengthened. Table 1, provides an overview of the processes that precede polyphenol extraction. Only TPC is given as extract quality. I think that something more should be said about the quality of the extracts obtained, not only in terms of tests, but also something more about the amount and composition of polyphenols.
I believe that with this appendix the paper should be accepted for publication.
Reviewer 2 Report
Congratulations to the author. The review is very well written and has Scientific Soundness. I suggest minor revisions
In line 142: “1 ms-1”, change to 1 m/s and standardize the review.
I suggest making a table with the main compounds found in the olive leaves or the main classes of phenolics found.
Thus, it is possible to describe the effect of different classes of compounds in each process: drying, size reduction, and blanching. If to exist in the literature, of course.
In section 2.1 Drying, differences in the concentrations of oleuropein (secoiridoid) and caffeic acid (phenolic acid) are described. However, classes of compounds are different, and they have different behaviours.
